# Performance Analysis of the C/C Composite Cylindrical Reverse Inter-Shaft Gas Film Seal

**Hua Su \*, Shuaike Zhao and Xiaofei Yang**

School of Mechanical Engineering, Northwestern Polytechnical University, Xi'an 710072, China
\* Correspondence: huasu@nwpu.edu.cn

**Abstract:** The reverse inter-shaft gas film seal is a gas path seal used in the intermediate bearing cavity of an aero-engine, which is challenging to implement due to its special installation and usage conditions. This paper proposes a C/C (Carbon/Carbon) composite cylindrical reverse inter-shaft seal structure and carries out a performance simulation analysis based on bidirectional fluid–solid coupling technology. The results show that the cylindrical reverse inter-shaft gas film seal with double-layer C/C composite sealing rings with different material mesoscopic parameters can balance seal leakage and friction power consumption and is beneficial to improving the comprehensive performance of the seal. As the mesoscopic parameters of the inner sealing ring material increase in warp and weft density, the sealing leakage rate decreases, and the gas film force and gas friction power consumption all increase. As the pressure difference increases, the sealing leakage rate and gas film force increase. As the rotation speed of the inner and outer rotors increases, the seal leakage rate increases, and the gas film force decreases. A C/C composite sealing ring cylindrical reverse inter-shaft gas film seal has a lower leakage rate and larger gas film force than a graphite sealing ring cylindrical reverse inter-shaft seal, which confers certain performance advantages. The work in this paper can provide a reference for the design of reverse inter-shaft seals.

**Keywords:** inter-shaft gas seal; fluid–structure coupling; C/C composite material; leakage; gas film force; friction power consumption

## 1. Introduction

The ongoing need for higher-performance industrial turbines has led to extensive efforts to improve various components of gas turbines [1,2]. To meet the thrust-to-weight ratio and cost of ownership goals of advanced engines, rotor system technology development has been directed toward counterrotating rotors with a straddle-mounted high rotor coupled to the low rotor through an inter-shaft bearing [3]. Therefore, intermediate bearings are required for support between the high- and low-pressure turbines that rotate on the same axis and the connecting shafts of their driven compressors and fans. The sealing of the gas flow passages in the intermediate bearing cavities between the two rotating shafts is inter-shaft sealing. To improve performance, some engines use two engine rotors with opposite rotation directions, resulting in multiple increases in the speed of this type of inter-shaft seal (referred to as a reverse inter-shaft seal), with a surface relative speed exceeding 300 m/s. In addition, in this sealing area, axial movement on the order of 0.30 inches between the shafts must be tolerated [4]; the small installation space and difficult operation and maintenance make it one of the most difficult sealing applications.

There are two main types of sealing technologies that have been used for inter-shaft seals for aero-engines to the present time, including labyrinth and graphite (carbon) seals. Due to the high sliding speed of the reverse rotation inter-shaft seal, it is only possible to use multi-stage labyrinth seals. Labyrinth seals are non-contacting seals with high leakage flows, resulting in high lubricant consumption [4].

To reduce seal leakage and heat generation, since the 1980s, the graphite face gas film sealing technology for inter-shaft seals has been used with significant progress [5]. The

non-contacting high-speed (800 ft/s) seal incorporating hydrodynamic lift geometry with spiral grooves in the seal plates was demonstrated through a series of rig tests. It was found that the leakage was one-third of the labyrinth seal operating under similar conditions [3].

Based on this graphite face gas film sealing structure, Wang et al. analyzed the performance of the face inter-shaft seals [6,7]. It was found that the hydrodynamic force can be generated under a certain stable running state and reach a non-contact state. Although the face gas seal technology has been the basis of many applications and achievements in dry gas seals, mechanical seals, and other ground rotating machinery [8–10], the currently studied face gas film seal is mainly used between the stator and rotor with relatively stable operating conditions and is not suitable for large axial displacement (about 5 to 6 mm) conditions. It is critical to control the gas film clearance between the two rotating faces, especially when the rotor has axial movement. Studies have shown that [11], although the leakage rate of the currently used graphite ring inter-shaft gas film seal is significantly less than that of the labyrinth seal, the wear is serious, and smoke phenomena have occurred during tests, which cannot meet the working requirements of the inter-shaft seal. For this reason, researchers have put forward a lot of improvement schemes for the structure of inter-shaft seals. It also can be seen from some face inter-shaft seal design patents published in recent decades [12–14] that design ideas address aspects such as gas path structure improvement, end surface structure design, and adjustment compensation structure. However, these patents have no basis in substantive research.

In order to solve the problem of inter-shaft sealing end face collision caused by large axial displacement of the rotor, cylindrical gas film technology has been proposed for reverse inter-shaft sealing. Hou et al. [15,16] proposed a cylindrical reverse inter-shaft gas film seal with an elastic support structure and used an aeroelastic coupling calculation method to analyze the effects of rotor rotation direction, seal ring width, rotor speed, and rotor radius on sealing performance. However, the cylindrical reverse inter-shaft sealing structure of the elastic support structure is complex, making it difficult to process, install, and adjust. Hou et al. proposed a flexible cylindrical seal structure with metal rubber [17] and used numerical methods to calculate the effects of rotor tilt, centrifugal expansion, and other factors on the gas film pressure distribution under stable conditions. Zhao et al. [18] analyzed the performance of graphite ring cylindrical reverse inter-shaft seals in different lubrication states and found that centrifugal expansion and rotational speed of the seal ring have a significant impact on the sealing performance.

The cylindrical inter-shaft seal structure proposed by Zhao et al. [18] is similar to the floating ring seal structure. There has been a lot of research on the performance of floating ring seals. Arghir M. [19] presented an analytic model which is able to take into account only the synchronous periodic whirl motion of the floating ring. Additionally, more researchers studied the performance of floating ring seals through experiments [20–23]. The above research studies provided a reference for the research of inter-shaft seals. However, the operating conditions of the floating ring seal and inter-shaft seal are different. The sealing ring of the floating ring seal is fixed, which belongs to the stator–rotor seal. However, the sealing ring of the cylindrical inter-shaft seal studied in this paper is rotary, belonging to the rotor–rotor seal. The centrifugal expansion caused by the rotation of the sealing ring will also have a great impact on the sealing performance. Moreover, the hydrodynamic force effect is reduced under the reverse rotating condition, which increases the risk of collision between the sealing ring and the inner rotor.

For the cylindrical inter-shaft seal, it is difficult for a single graphite seal ring to reduce leakage, heat loss, and wear. Based on the high strength and designability of C/C composite material, a double-layer sealing ring structure is proposed in this work, and the mechanical properties of the two layers are obtained by using different mesoscopic parameters. In this way, both centrifugal expansion (the outer rotor is made of materials with higher stiffness) and contact wear (the inner rotor is made of materials with lower stiffness) can be reduced, which is expected to achieve better overall performance.

According to the service conditions of the cylindrical inter-shaft seal, the sealing ring material should have good thermal performance, low and stable friction coefficient, high strength, and low weight. Considering the self-lubricating characteristics of C/C composites, it helps to reduce seal friction and wear, and compared to graphite sealing materials, it shows significant improvement in strength, friction and wear, and heat dissipation capacity [24,25]. Hence, we propose a layered C/C composite cylindrical reverse inter-shaft seal, establish a fluid–solid coupling performance analysis model, study its leakage and bearing capacity under different operating conditions, and compare its performance with that of the graphite cylindrical reverse inter-shaft seal. This paper provides a new idea for the design of cylindrical reverse inter-shaft seals.

## 2. Cylindrical Inter-Shaft Sealing Structure and Materials

As shown in Figure 1, the structural composition of the cylindrical reverse inter-shaft seal consists of an outer rotor (4), snap ring (5), anti-rotation pin (3), sealing ring (1 and 2), sealing seat (6), and inner rotor (8). The seal seat (6) has a fixed connection to the outer rotor (4) through a snap ring (5), and it rotates with the outer rotor (4). There is a relative eccentricity between the sealing ring and inner rotor shaft due to the fit tolerance between them, so a convergent wedge-shaped space can be formed between them. As the inner and outer rotors rotate in a wedge-shaped space, a hydrodynamic force generated by the gas film, the centrifugal force generated by the rotation of the sealing ring, and the friction force between the sealing ring and the end face of the outer rotor can be balanced in the radial direction, ensuring the floating of the sealing ring, satisfying the non-contact operation requirements of the cylindrical reverse inter-shaft gas film seal.

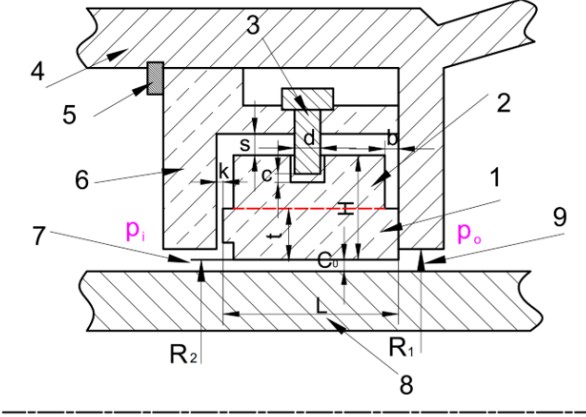

1—Inner sealing ring; 2—outer sealing ring; 3—anti-rotation pin;
4—outer rotor; 5—snap ring; 6—seal seat; 7—pressure inlet;
8—inner rotor; and 9—pressure outlet.

**Figure 1.** Structural composition of cylindrical reverse shaft seal.

In the C/C composite cylindrical reverse inter-shaft gas film seal, the high-speed rotation of the sealing ring can cause centrifugal expansion deformation, and the radial runout of the rotor can cause severe wear due to the collision between the inner rotor and the inner surface of the sealing ring. To reduce the centrifugal expansion deformation and wear of the sealing ring, it is divided into inner and outer layers that use C/C composite materials with different elastic moduli. The 2.5D C/C woven composite material is selected as sealing ring, and the material model is shown in Figure 2. The outer layer has respective warp and weft densities of 32 roots/cm and 16 roots/cm, and the inner layer has warp and weft densities of 22 roots/cm and 14 roots/cm, respectively. The average stiffness method is used to calculate mechanical property parameters, such as the equivalent elastic modulus of the C/C composite material under different microscopic parameters [19]. The structural

parameters of cylindrical reverse inter-shaft gas film seal studied in this article are shown in Table 1. The material properties are shown in Table 2.

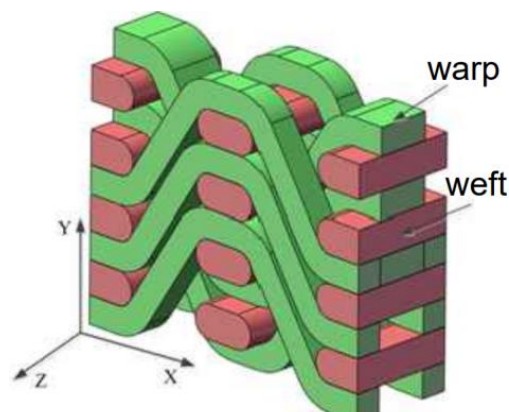

**Figure 2.** A 2.5D composite material model.

**Table 1.** Geometric parameters of C/C composite cylindrical reverse axial gas film seal structure.

| Parameter | Value |
|---|---|
| Outer rotor inner diameter $R_1$ | 77.7 mm |
| Inner diameter of sealing ring $R_2$ | 77.2 mm |
| Radial height of sealing ring $H$ | 5 mm |
| Axial width of sealing ring $L$ | 11.9 mm (8–23) |
| Inner layer height $t$ | 1.8 mm (0.5–5) |
| Sealing dam width $b$ | 0.8 mm |
| Axial clearance between sealing ring and sealing seat $k$ | 0.8 mm |
| Initial installation clearance between sealing ring and inner rotor $C_0$ | 0.03 mm |
| Diameter of anti-rotation pin $d$ | 2 mm |
| Clearance between anti-rotation pin and rotor $c$ | 0.5 mm |
| Clearance between OD of floating ring and seal set $s$ | 3 mm |

**Table 2.** Properties of inter-shaft seal materials.

| | $E_x$ (GPa) | $E_y$ (GPa) | $E_z$ (GPa) | $v_{xy}$ | $v_{xz}$ | $v_{zy}$ | $G_{yz}$ (GPa) | $G_{xz}$ (GPa) | $G_{xy}$ (GPa) |
|---|---|---|---|---|---|---|---|---|---|
| Inner layer of sealing ring | 19.1 | 20.1 | 52.8 | 0.4 | 0.09 | 0.09 | 6.3 | 6.3 | 10 |
| Outer layer of sealing ring | 37.1 | 47.4 | 68.2 | 0.2 | 0.14 | 0.13 | 10.6 | 10.2 | 12.1 |
| Inner and outer rotors | | 209 | | | 0.34 | | | 79 | |

## 3. Force Analysis of Cylindrical Inter-Shaft Seal

Figure 2 shows the force diagram of the C/C composite cylindrical reverse inter-shaft gas film seal structure. Taking the sealing ring as the analysis object, when the seal works, the resultant forces exerted by the fluid on the sealing ring are $F_1$, $F_3$, and $F$. $F_1$ and $F_3$ are approximately equal to the inlet pressure multiplied by their acting area, and $F$ is the hydrodynamic force formed by the gas film gap between the sealing ring and inner rotor. The centrifugal force generated by the rotation of the outer rotor driven by the anti-rotation pin to rotate the sealing ring is $F_2$, the friction force between the sealing ring and outer rotor is $F_f$, and the reaction force of the outer rotor against the sealing ring is $F_4$. Under

the combined action of forces $F_1$, $F_2$, $F_3$, $F_4$, $F$, and $F_f$, the sealing ring is in a radial force equilibrium state.

### 3.1. Friction $F_f$

Following Figure 3, the normal force acting on the nose of the sealing ring is

$$F_4 = p_i \pi [(R_2 + t)^2 - R_2{}^2] - p_o \pi (R_1{}^2 - R_2{}^2) \tag{1}$$

and will be balanced by the asperity contact forces $F_{4,asp}$ and by a hydrostatic effect $F_{4,fluid}$

$$F_4 = F_{4,fluid} + F_{4,asp} \tag{2}$$

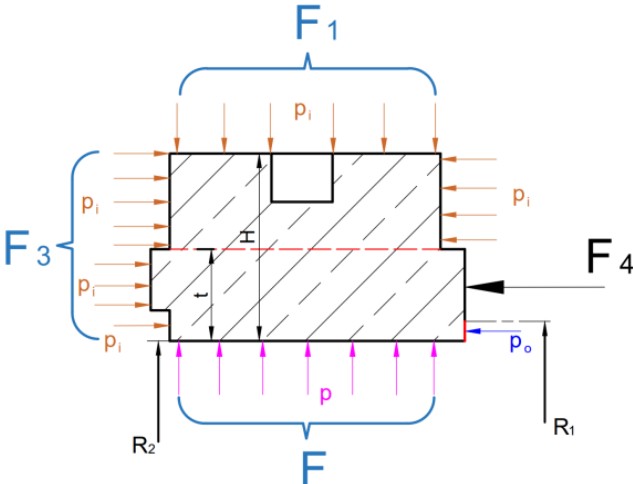

**Figure 3.** Force diagram of C/C composite cylindrical reverse inter-shaft gas film seal.

Then, the friction force is [23]

$$F_f = F_{4f,fluid} + F_{4f,asp} \approx F_{4f,asp} = f F_{4,asp} \tag{3}$$

The normal force $F_{4,asp}$ is estimated by considering the total number of elastic contacts between asperities.

$$F_{4,asp} = \frac{3}{4} N E' \beta^{0.5} \int_h^c \frac{35}{32c^7} (c^2 - \xi^2)^3 (\xi - h)^{3/2} d\xi \tag{4}$$

where $f$ is the friction coefficient between the sealing ring and outer rotor, $p_o$ is the approximate inlet pressure, $p_o$ is the outlet pressure, $N = (\eta_1 + \eta_2) S_c$ is the total number of asperities, $\eta_1$, $\eta_2$ are the density of asperities of the two contacting surfaces, $S_c = \pi (R_2 + t)^2 - \pi R_1{}^2$, $E'^{-1} = (1 - v_1{}^2) E_1{}^{-1} + (1 - v_2{}^2) E_2{}^{-1}$, $c = 3\sigma$, where $\sigma = \sqrt{\sigma_1{}^2 + \sigma_2{}^2}$ is the combined standard deviation of the two contacting surfaces, $\beta$ is the combined average radius of the spherical roughness, and $h$ is the distance between the flat surface and the mean plane of the equivalent rough surface. Here, $\eta_1$ is 2470 mm$^{-2}$, $\eta_2$ is 2282 mm$^{-2}$, $\sigma_1$ is 0.031 μm, $\sigma_2$ is 0.34 μm, and $\beta$ is 17.2 μm.

### 3.2. Gas Film Force of the Gas Film F

$$\overline{F} = F_x \overline{i} + F_y \overline{j} \tag{5}$$

$$F_x = \int_0^l \int_0^{2\pi} p R_2 \cos \theta d\theta dz$$
$$F_y = \int_0^l \int_0^{2\pi} p R_2 \sin \theta d\theta dz \tag{6}$$

where $l$ is the width of the sealing ring; $p$ is the gas film pressure; $R_2$ is the inner diameter of the sealing ring; and $\theta$ is the polar coordinate rotation angle.

*3.3. Centrifugal Force $F_2$*

$$F_2 = m\omega^2 e \tag{7}$$

where $m$ is the seal ring mass, $e$ is the seal ring eccentricity, and $\omega$ is the seal ring angular velocity.

When the radial force of the sealing ring is balanced,

$$\vec{F}_1 + \vec{F}_2 + \vec{F} + \vec{F}_f = 0 \tag{8}$$

## 4. Numerical Method for the Performance of Cylindrical Inter-Shaft Seals

*4.1. Governing Equations*

The gas film force and high-speed rotation of the seal ring cause the seal ring to deform, which in turn will affect the flow state of the seal gas, i.e., the interaction between the flow field and solid field, which is a typical two-way fluid–solid coupling problem, whose basic governing equations include those of fluid and solid fields. The fluid control equations include the conservation of mass, momentum, and energy.

The mass conservation equation for the gas in the gas film seal gap between C/C composite cylindrical reversal shafts can be expressed as

$$\frac{\partial \rho}{\partial t} + \frac{\partial(\rho u)}{\partial x} + \frac{\partial(\rho v)}{\partial y} + \frac{\partial(\rho w)}{\partial z} = 0 \tag{9}$$

where $\rho$ is the density of the sealing gas; $t$ is the time; and $\vec{u}$ is the velocity vector, with respective components $u$, $v$, and $w$ in the $x$, $y$, and $z$ directions.

The momentum conservation equations for the fluid in three directions in a rectangular coordinate system are

$$\frac{\partial(\rho u)}{\partial t} + div(\rho u \vec{u}) = -\frac{\partial p}{\partial x} + \frac{\partial \tau_{xx}}{\partial x} + \frac{\partial \tau_{yx}}{\partial y} + \frac{\partial \tau_{zx}}{\partial z} + F_x \tag{10}$$

$$\frac{\partial(\rho v)}{\partial t} + div(\rho v \vec{u}) = -\frac{\partial p}{\partial y} + \frac{\partial \tau_{xy}}{\partial x} + \frac{\partial \tau_{yy}}{\partial y} + \frac{\partial \tau_{zy}}{\partial z} + F_y \tag{11}$$

$$\frac{\partial(\rho w)}{\partial t} + div(\rho w \vec{u}) = -\frac{\partial p}{\partial z} + \frac{\partial \tau_{xz}}{\partial x} + \frac{\partial \tau_{yz}}{\partial y} + \frac{\partial \tau_{zz}}{\partial z} + F_z \tag{12}$$

where $div(\vec{a}) = \partial a_x/\partial x + \partial a_y/\partial y + \partial a_z/\partial z$; $p$ is the pressure on the surface of the fluid microelement; $\tau_{xx}$, $\tau_{xy}$, and $\tau_{xz}$ are the components of the viscous shear force $\tau$ generated by the fluid due to viscous action acting on the surface of the microelement; and $F_x$, $F_y$, and $F_z$ are the volumetric forces acting in three directions on the fluid microelement.

For gas density, the ideal gas law provides

$$\rho = \frac{P}{RT} \tag{13}$$

where $R$ is the specific gas constant and $T$ is the temperature of the gas.

The finite element expression of the dynamic control equation of the sealing ring is

$$[M]\{\ddot{x}\} + [C]\{\dot{x}\} + [K]\{x\} = \{F(t)\} \tag{14}$$

which, under stable operation without considering the influence of rotor excitation, can be simplified as

$$[K]\{x\} = \{F\} \tag{15}$$

where $[M]$ is the sealing ring mass matrix; $[C]$ is the sealing ring damping matrix; $[K]$ is the stiffness matrix of the sealing ring; $\{F\}$ is the force vector, including all external forces acting on the seal ring; $\{x\}$ is the displacement vector of the sealing ring; $\{\dot{x}\}$ is the velocity vector, which is the first derivative of the displacement vector with respect to time; and $\{\ddot{x}\}$ is the acceleration vector, which is the second derivative of the displacement vector with respect to time.

The analysis of fluid–solid coupling of C/C composite cylindrical reverse inter-shaft gas film seals must also meet the most basic conservation principle at the interface of fluid–solid coupling, i.e., the stress and displacement of fluid and solid at the coupling surface must be equal, i.e.,

$$\tau_f \cdot n_f = \tau_s \cdot n_s \tag{16}$$

$$d_f = d_s \tag{17}$$

where $\tau$ is the coupling surface stress, and $d$ is the displacement of the coupling surface.

*4.2. Performance Index*

The leakage rate, gas film force, and friction power consumption reflect the sealing performance of C/C composite cylindrical reverse inter-shaft gas film seals.

The mass flow rate of leakage is determined through the integration of flow flux around the computational domain as

$$Q = \iint \rho v d\sigma \tag{18}$$

where $Q$ is the density of the gas, $\rho$ is the density of the gas, $\sigma$ is the total peripheral area around the edges of the computational domain, and $v$ is the velocity at the boundary.

The film force is calculated in Equation (2), and the friction power consumption is

$$W = \iint \left| \vec{\tau_v} \right| \omega r dx dy \tag{19}$$

where $\omega$ is the relative angular velocity between the sealing interfaces, and $\vec{\tau_v}$ is the vector sum of the viscous shear force of the gas in the circumferential and axial directions,

$$\vec{\tau_v} = \tau_d \hat{d} + \tau_\varphi \hat{\varphi} \tag{20}$$

where $\tau_d$ and $\tau_\varphi$ are the respective viscous shear forces in the axial and circumferential directions.

*4.3. Method of Solution*

4.3.1. Numerical Analysis Method of the Flow Field

According to the geometric dimensions of the flow field of the C/C composite cylindrical reverse inter-shaft gas film seal, the initially established seal flow field analysis model is shown in Figure 4, where $\Phi1$ is the inner diameter of the seal ring, $\Phi2$ is the outer diameter of the inner rotor, and $e$ and $\theta$ are, respectively, the eccentricity and eccentricity angle between the seal ring and inner rotor.

The mesh generation of the flow field model of the C/C composite cylindrical reverse inter-shaft gas film seal is performed in the MESH software that comes with the ANSYS WORKBENCH platform. Due to the thin gas film with a thickness of micrometers, it is difficult to accurately reflect the flow field characteristics if there are too few grid layers in the gas film thickness direction, while too many grid layers will increase the calculation cost. Therefore, it is necessary to verify the grid independence of the flow field. The flow field grid is divided into 4, 5, 6, 7, and 8 layers in the film thickness direction. Table 3 shows the sealing gas film force and leakage rate corresponding to these five grid layers. Based on the calculation results, the gas film grid is determined to be divided in the thickness direction into six layers, for a total of 822,900 grids. Figure 4b shows a grid model of the flow field.

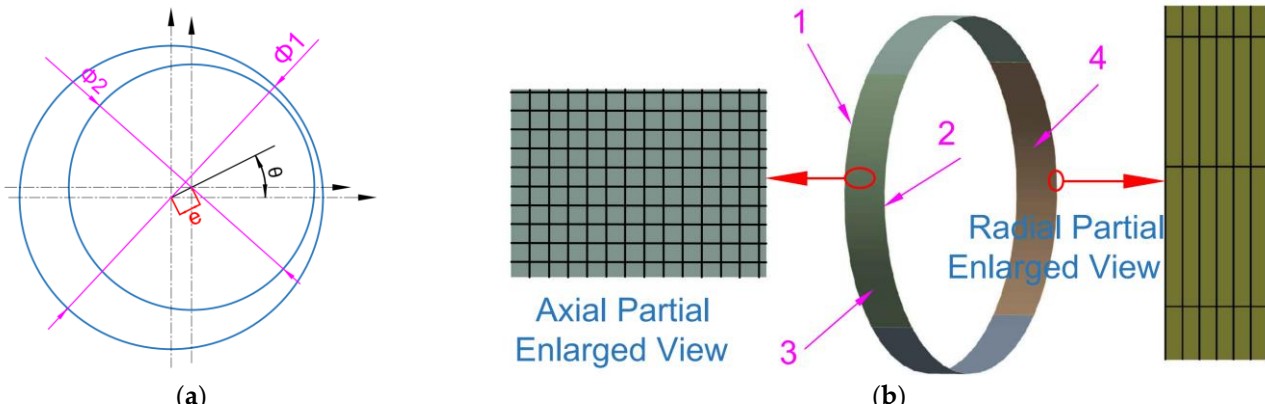

**Figure 4.** Analytical model for flow field of C/C composite cylindrical reverse inter-shaft gas film seal: (**a**) 2D plan; (**b**) grid chart.

**Table 3.** Sealing performance and relative error under different grid layers. (The inlet pressure difference is 80 kPa, the internal rotor speed is 12,000 rpm, the outer rotor speed is 9000 rpm, rotation of inner and outer rotors is reversed, and the relative velocity of surface motion is 170.24 m/s).

| Number of Grid Layers | Total Number of Grids | Gas Film Force $F$ (N) | Leakage $Q$ (g/s) |
|:---:|:---:|:---:|:---:|
| 4 | 548,568 | 7.877 | 3.3989 |
| 5 | 685,710 | 7.9237 | 3.2757 |
| 6 | 822,852 | 7.9586 | 3.2103 |
| 7 | 959,994 | 7.9878 | 3.1683 |
| 8 | 1,097,136 | 7.9731 | 3.1423 |
| Maximum relative error | | 1.2% | 7.5% |

In the flow field analysis of C/C composite cylindrical reverse inter-shaft gas film seals, the flow field boundary conditions are set as follows (as shown in Figure 4b)

1. The fluid inlet end face (position 1) is set as the pressure inlet, and the pressure value is the intake pressure on the sealing ring;
2. The fluid outlet end face (position 2) is set as the pressure outlet, and the pressure value is the low-pressure chamber pressure (atmospheric pressure);
3. The outer wall surface (position 3), where the fluid contacts the sealing ring, is set as a sliding free rotating wall surface, and the rotational speed is that of the outer rotor;
4. The inner wall surface (position 4), where the fluid contacts the inner rotor, is set as a non-sliding rotating wall surface whose rotational speed is that of the inner rotor.

After calculation by formula (21), the maximum Reynolds number under this working condition is about 794, and the fluid mainly presents a laminar flow state. Therefore, we select a laminar flow model for simulation analysis of the convection field.

$$Re = \frac{\rho r \omega h_o}{\mu} \tag{21}$$

where $\rho$ is the gas density; $r$ is the middle radius value of the sealed fluid domain ($r = R_2 - \frac{C_0}{2}$); $\omega = \frac{\omega_1 + \omega_2}{2}$ ($\omega_1$, $\omega_2$ represents the angular velocity of the inner rotor and the outer rotor, respectively); $h$ is the thickness of the gas film; and $\mu$ is the fluid dynamic viscosity.

4.3.2. Numerical Analysis Method of the Solid Field

The seal solid field analysis model and grid division are shown in Figure 5. Using the multi-zone grid division method, considering the calculation cost and accuracy issues after

a large number of trial calculations, the seal ring grid size is taken as 0.5 mm, and the outer rotor grid size as 1.1 mm. The final number of solid field grids is 270,000.

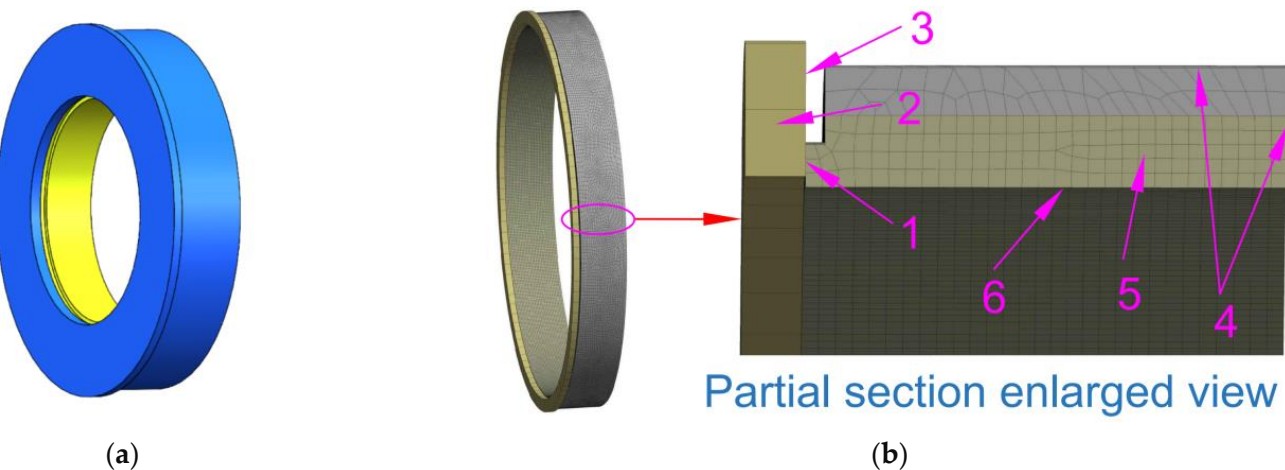

**Figure 5.** Grid division of solid field for C/C composite cylindrical reverse inter-shaft gas film seal: (**a**) solid field geometric model; (**b**) mesh division (with boundary conditions).

The boundary conditions for solid field analysis of C/C composite cylindrical reverse inter-shaft gas film seals are applied as shown in Figure 5b:

1. The contact surface between the outer rotor and sealing ring (position 1) is set as a friction contact pair with a friction coefficient of 0.2 [26];
2. The outer rotor (position 2) and sealing ring (position 5) are set with a rotational speed around the central shaft, which is that of the outer rotor;
3. Constraints are applied to the outer rotor surface (position 3), boundary conditions are set that allow the outer rotor to only rotate and not move axially or radially, and the motion of the outer rotor is simulated;
4. The gas flows through the surface area of the sealing ring (position 4) and applies fluid inlet pressure;
5. The gas flows through the inner surface area of the sealing ring (position 6), which is set as a fluid–solid coupling surface.

### 4.4. Analysis Process

The flowchart for analyzing the performance of C/C composite cylindrical reverse inter-shaft gas film seals based on fluid–solid coupling is shown in Figure 6.

### 4.5. Verification of the Numerical Method

To validate the proposed analytical method for cylindrical reverse inter-shaft gas film seals based on fluid–solid coupling, the calculation results are compared with experimental and simulation results in the literature [18]. The test sealing ring material is graphite (elastic modulus 18 GPa, Poisson's ratio 0.3, and density 1.8 g/cm$^3$). The different rotational speed combinations of the inner and outer rotors are shown in Table 4.

Figure 7 compares the leakage rate calculated by this simulation and the reference results under normal temperature, gas inlet pressure of 0.12 MPa, outlet atmospheric pressure, and different rotational speed combinations. From the figure, it can be seen that the simulation results of the leakage rate in this article are consistent with the curve change trend of the test leakage rate results. The maximum relative error between the simulation results of the leakage rate and the test results does not exceed 11.1%. Moreover, the calculation results are closer to the test values than those obtained by using the average Reynolds equation in the reference, which validates the analysis method.

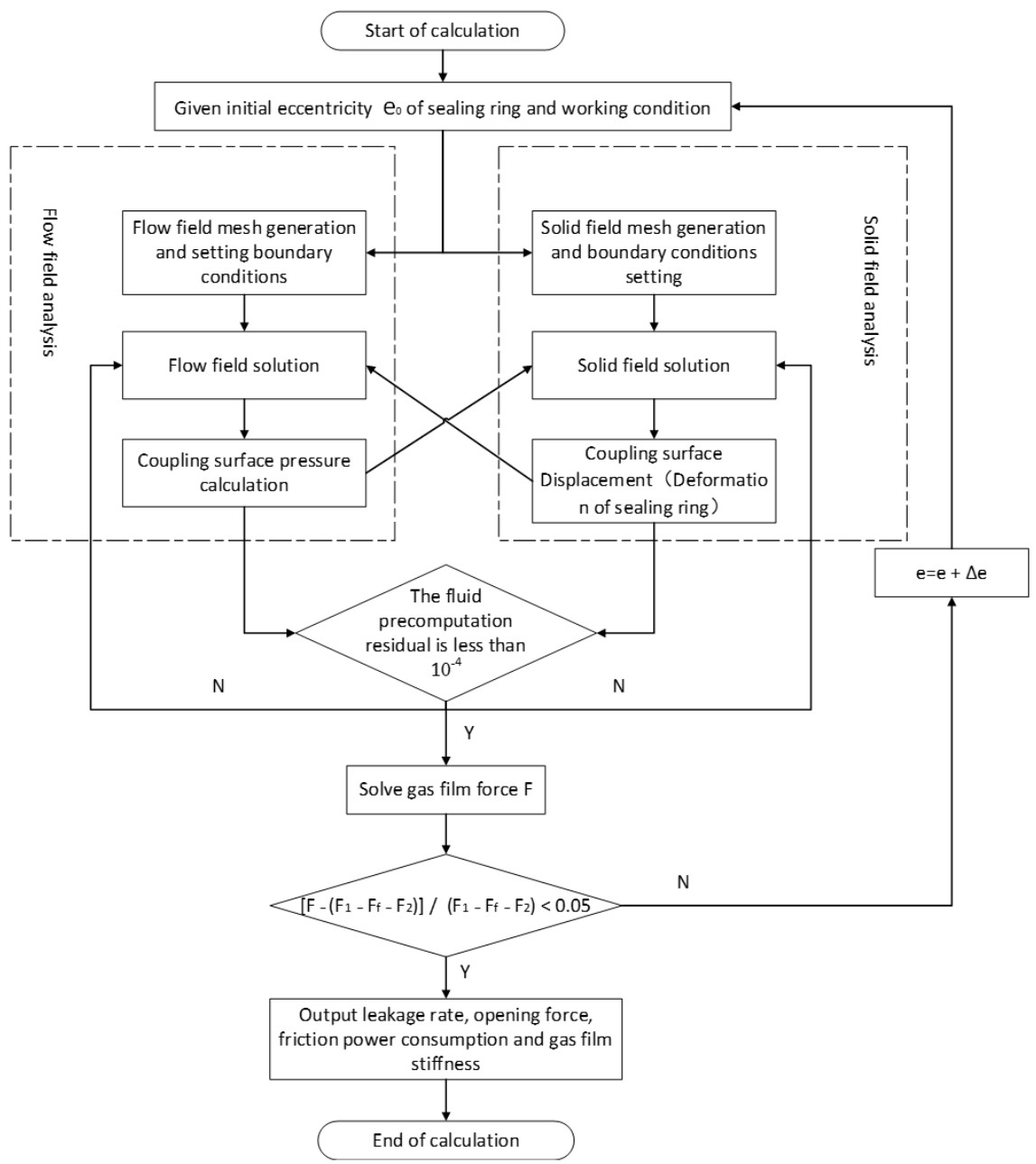

**Figure 6.** Fluid–solid coupling analysis process.

**Table 4.** Speed group of inner and outer rotors (reverse rotation of inner and outer rotors) [18].

| Speed Group $N_S$ | Outer Rotor Speed (rpm) | Inner Rotor Speed (rpm) | Relative Surface Speed (m/s) |
|:---:|:---:|:---:|:---:|
| 1 | 5000 | 8000 | 105.36 |
| 2 | 7000 | 10,000 | 137.80 |
| 3 | 9000 | 12,000 | 170.24 |
| 4 | 10,000 | 14,000 | 194.55 |
| 5 | 11,000 | 15,000 | 210.77 |
| 6 | 12,000 | 16,000 | 226.99 |

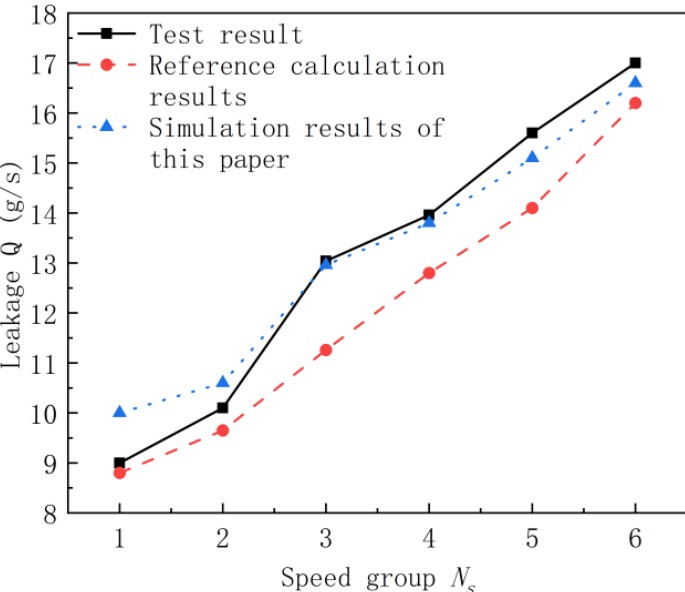

**Figure 7.** Comparison of leakage rates simulated in this paper and reference [18] under different rotational speed combinations of inner and outer rotors.

## 5. Analysis Results

Figures 8 and 9 show the gas flow field pressure distribution, film thickness distribution, and velocity under the pressure difference of 80 kPa, the inner rotor speed of 12,000 rpm, and the outer rotor speed of 9000 rpm. The seal ring deformation is shown in Figure 10. It can be seen from Figure 8 that the maximum pressure occurs near the thinnest gas film thickness due to the effect of sealing convergence clearance on flow obstruction.

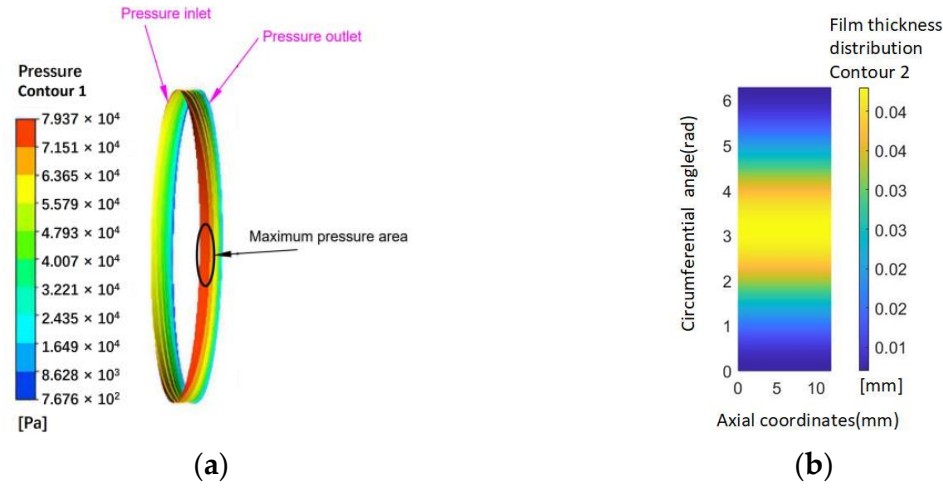

(**a**)         (**b**)

**Figure 8.** Axial cloud map distribution: (**a**) axial pressure distribution cloud diagram; (**b**) axial film thickness distribution cloud map (unfold along the minimum film thickness value (the maximum pressure value)).

As can be seen from Figure 9, the flow lines are chaotic, and there is a certain angle between the flow lines and the axial direction of the sealing ring, and the flow lines are less distributed in the position where the gas film is thin. This is mainly because the interface contacting the sealing ring and the inner rotor in the flow field will produce shear force. Under the combined action of shear force and pressure difference in the flow field, the flow direction of the gas will be deflected. In addition, the reverse rotation of the sealing ring and the inner rotor makes the shear force generated by the interface in contact with the sealing ring and the inner rotor go in the opposite direction, resulting in the reverse flow of

the inner and outer fluid in the circumferential direction. Therefore, the streamline of the entire flow field shows a relatively chaotic distribution.

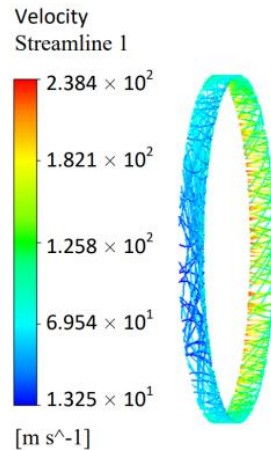

**Figure 9.** Velocity streamline diagram.

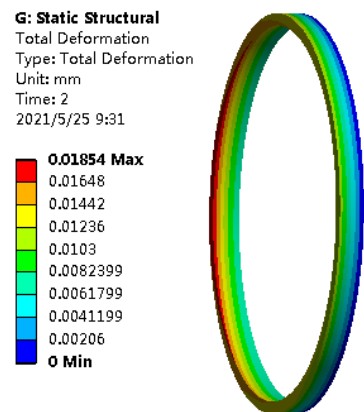

**Figure 10.** Deformation of the seal ring.

As shown in Figure 10, the sealing ring deforms greatly near the entrance (left) and where the pressure is high. In order to ensure the convergence of finite element analysis, the right edge of the seal ring is treated as a fixed constraint.

*5.1. Effect of the Relative Height of the Inner Sealing Ring on Sealing Performance*

When the pressure difference is 80 kPa, the outer rotor speed is 9000 rpm, and the inner rotor speed is 12,000 rpm. Figure 11 shows the effect of the relative thickness of the inner layer of the sealing ring (=t/H) on the performance of the C/C composite cylindrical reverse inter-shaft gas film seal. Because the elastic modulus of the inner sealing ring material is less than that of the outer sealing ring, as the relative thickness of the inner layer of the sealing ring increases, the overall elastic modulus of the sealing ring decreases, and the centrifugal expansion increases, resulting in an increase in the gap between the sealing ring and the inner rotor, a decrease in the hydrodynamic pressure effect, and an increase in the leakage rate. There are slight fluctuations in gas film force, and the overall trend is gradually decreasing. However, the change rates of the leakage rate and the gas film force are low, with a maximum of approximately 1.1%, indicating that changing the relative thickness of the inner layer of the sealing ring has no significant impact on sealing performance and, under relatively stable operating conditions, has little impact on the sealing performance. Considering that the radial runout of the inner and outer rotors may cause the inner surface of the sealing ring to collide with the inner rotor, the thickness of the inner layer of the sealing ring can be appropriately increased to cushion their impact

collision. In addition, the weakening of the hydrodynamic pressure effect of the sealing gas reduces the viscous shear force inside the fluid, so the frictional power consumption of the sealing gas decreases with the increase in the relative thickness of the inner layer of the sealing ring.

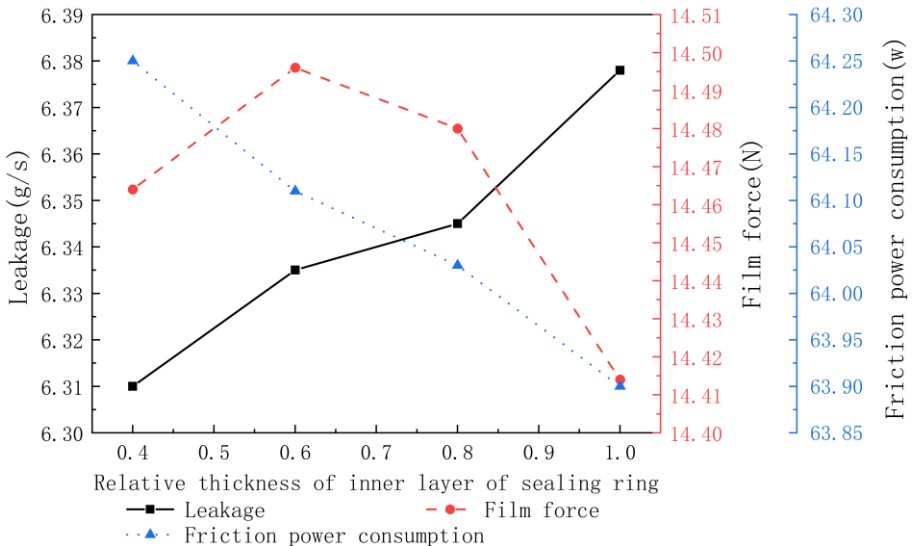

**Figure 11.** Effect of relative height of inner sealing ring on sealing performance.

From the above analysis, it can be seen that the change in the relative height of the inner layer reflects the change in the overall elastic modulus of the sealing ring, which is manifested in the contradictory trend of leakage and friction power consumption. Using a single-layer sealing ring structure with a small or large elastic modulus will, respectively, cause an increase in leakage or friction power consumption. To achieve a balance between the two, the numerical value corresponding to the intersection point of the curves of leakage and friction power consumption can be taken as the relative height of the inner layer of the sealing ring in Figure 11. Therefore, adopting C/C material seal rings with layered structures of different elastic moduli is beneficial for improving the comprehensive performance of the seal.

### 5.2. Effect of Mesoscopic Parameters of Composite Materials on Sealing Performance

Since the deformation of the inner layer of the sealing ring directly affects the sealing clearance, we analyze the impact of changes in the mesoscopic parameters of the inner layer sealing ring material on the sealing performance, while the material parameters of the outer rotor remain unchanged. The mesoscopic parameters of C/C materials include the density of the warp fiber bundles arranged in the composite material, i.e., the warp density (root/cm), and the density of the weft fiber bundles arranged in the composite material, i.e., the weft density (root/cm).

The calculated working conditions are as follows: an outer rotor speed of 9000 rpm, an inner rotor speed of 12,000 rpm, an inlet pressure difference of 80 kPa, and sealing structure parameters unchanged.

#### 5.2.1. Effect of Warp Density on Sealing Performance

Figure 12 shows the effect of warp density on the equivalent elastic property parameters of C/C composites. It can be seen that, with the increase in warp density, the elastic moduli E(x) in the warp direction and E(y) in the composite cell thickness direction both increase. Figure 13 shows the effect of the micro-weaving parameters of C/C composites on the sealing performance of the C/C composite cylindrical reverse inter-shaft gas film seal, from which it can be seen that, with the increase in warp density, the seal leakage rate decreases, while the gas film force and friction power consumption increase approximately

linearly. Under certain operating conditions, the increase in warp density increases the elastic modulus of the C/C composite material, resulting in a decrease in the centrifugal expansion deformation of the sealing ring, which reduces the sealing gap and, therefore, the sealing leakage rate. The reduction in the sealing gap will further compress the sealing gas, enhancing its hydrodynamic pressure effect, so the gas film force will increase. Reducing the sealing gap will increase the internal viscous shear force of the gas, so the frictional power consumption of the sealing gas will increase.

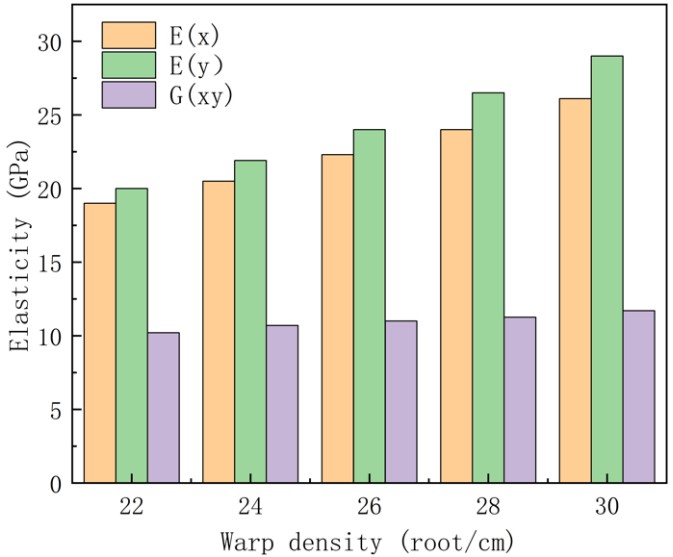

**Figure 12.** Effect of warp density on elastic performance parameters.

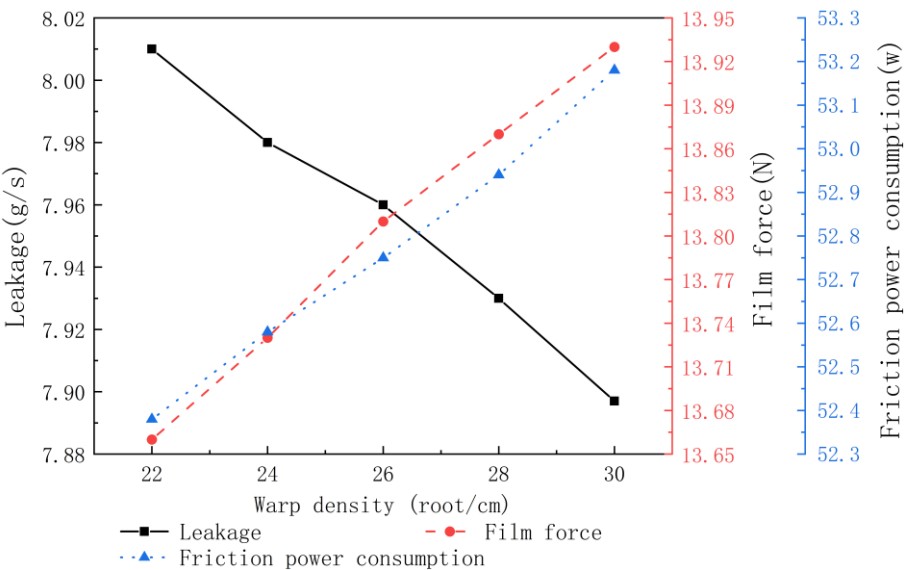

**Figure 13.** Effect of warp density on sealing performance.

## 5.2.2. Effect of Weft Density on Sealing Performance

Figure 14 shows the effect of weft density on the equivalent elastic property parameters of C/C composites. It can be seen that with its increase, the equivalent elastic modulus E(x) in the warp direction first decreases and then increases, while the elastic modulus E(y) in the thickness direction of the unit cell of C/C composites presents a gradually increasing trend. This is because, with the increase in weft density, the bending degree of the warp yarn increases, leading to the decrease in the elastic modulus E(x) in the warp direction. When the elastic modulus E(y) in the thickness direction of the unit cell increases,

and the weft density continues to increase, the increase in the warp curvature decreases, and the volume fraction of the warp in the entire composite increases, which leads to a trend of increasing the elastic modulus E(x) in the warp direction. When the weft density increases from 4 to 10 roots/cm, the elastic modulus E(x) in the warp direction decreases, and the elastic modulus E(y) in the composite cell thickness direction increases, but the overall stiffness of the inner sealing ring increases. When the weft density increases from 10 to 14 roots/cm, the elastic modulus E(x) in the warp direction increases, and the elastic modulus E(y) in the thickness direction of the composite unit cell increases. Therefore, as shown in Figure 15, as the weft density increases, the overall stiffness of the sealing ring increases, and the expansion deformation of the sealing ring decreases, i.e., the sealing gap decreases, so the sealing leakage rate gradually decreases. The reduction in the sealing gap will further compress the sealing gas and enhance its hydrodynamic pressure effect, so the gas film force will increase. Reducing the sealing gap will increase the viscous shear force inside the gas, so the gas friction power consumption will increase.

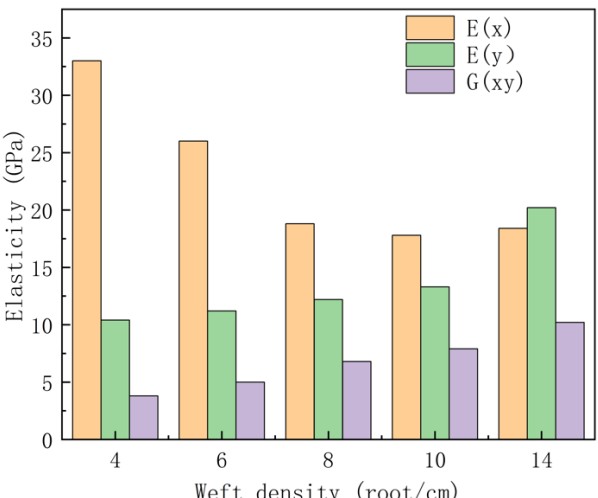

**Figure 14.** Effect of weft density on elastic performance parameters.

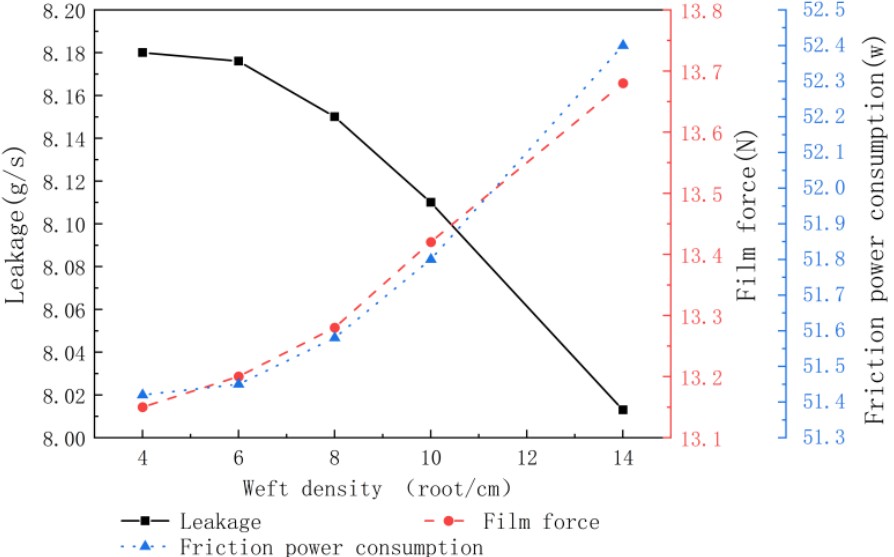

**Figure 15.** Effect of weft density on sealing performance.

*5.3. Comparison of the Gas Film Sealing Performance between C/C Composite Sealing Ring and Graphite Sealing Ring with Cylindrical Reverse Shaft*

Figure 16 shows the effect of inlet pressure differences on the leakage rate and gas film force of the cylindrical reverse inter-shaft gas film seal of the C/C composite seal ring and graphite seal ring when the inner rotor speed is 16,000 rpm, and the outer rotor speed is 12,000 rpm. As the inlet pressure difference increases, the leakage rate of the cylindrical reverse inter-shaft gas film seal of the C/C composite seal ring and graphite seal ring increases. This is because the increase in the inlet pressure difference increases the axial flow ability of the seal gas, so the leakage rate increases. However, the leakage rate of the C/C composite material cylindrical reverse axial gas film seal is lower because the C/C composite material used for the outer sealing ring has a large elastic modulus, which will hinder the centrifugal expansion of the inner ring during high-speed rotation, and because the sealing gap is smaller than that of the graphite sealing ring. Therefore, the leakage rate of the C/C composite material sealing ring's cylindrical reverse inter-shaft gas film seal is less than that of the graphite sealing ring.

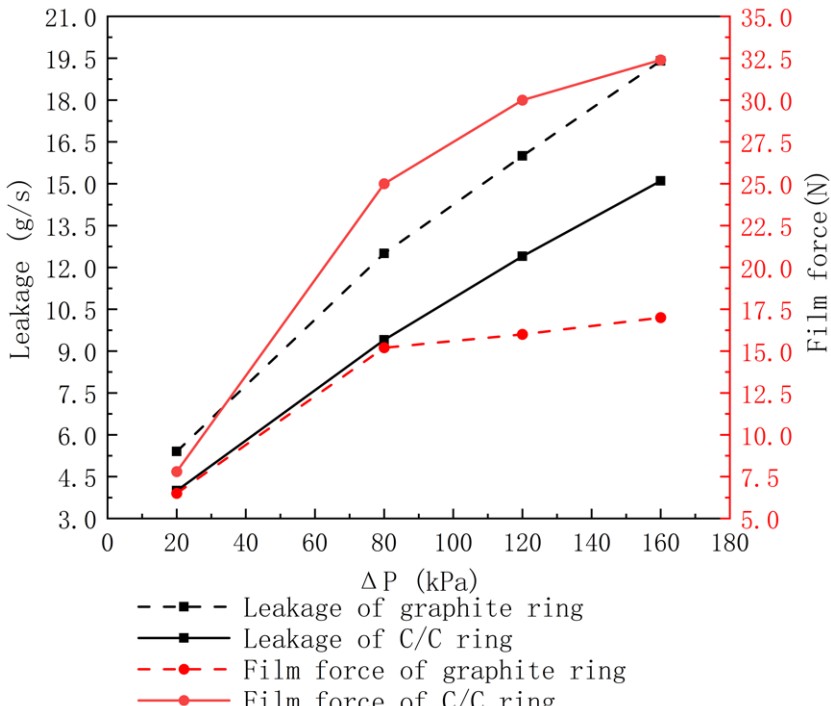

**Figure 16.** Comparison of the influence of pressure difference on the sealing performance of two materials.

As the inlet pressure difference increases, the gas film force of the cylindrical reverse inter-shaft gas film seal of both the C/C composite seal ring and graphite seal ring increases, because the increase in the inlet pressure difference enhances the hydrodynamic pressure effect of the sealing gas, increasing the gas film force. The gas film force of the cylindrical reverse inter-shaft gas film seal of the C/C composite seal ring is greater than that of the graphite seal ring, for the same reason that the leakage rate of the cylindrical reverse inter-shaft gas film seal of the C/C composite seal ring is less than that of the graphite seal ring. That is, the centrifugal expansion deformation of the C/C composite sealing ring is small, and the sealing gap is smaller than that of the graphite ring, making the sealing gas fluid dynamic pressure effect stronger, so the gas film force is greater than that of the graphite ring.

Figure 17 shows the effect of different combinations of inner and outer rotor speeds on the leakage rate and gas film force of the cylindrical reverse inter-shaft gas film seal of the C/C composite seal ring and graphite seal ring when the inlet pressure difference is 160 kPa. The speed combinations of the inner and outer rotors involved in the analysis

are shown in Table 5. As the rotational speed of the rotor increases, the leakage rate of the cylindrical reverse inter-shaft gas film seal of the C/C composite seal ring and the graphite seal ring increases because of the increase in the rotational speed of the rotor, which increases the centrifugal expansion deformation of the seal ring and the sealing gap, so the leakage rate increases. In addition, the increase in rotor speed changes the direction of gas flow, which also has a certain impact on leakage. Since the centrifugal expansion deformation of the C/C composite sealing ring is less than that of the graphite ring, the leakage rate is also lower.

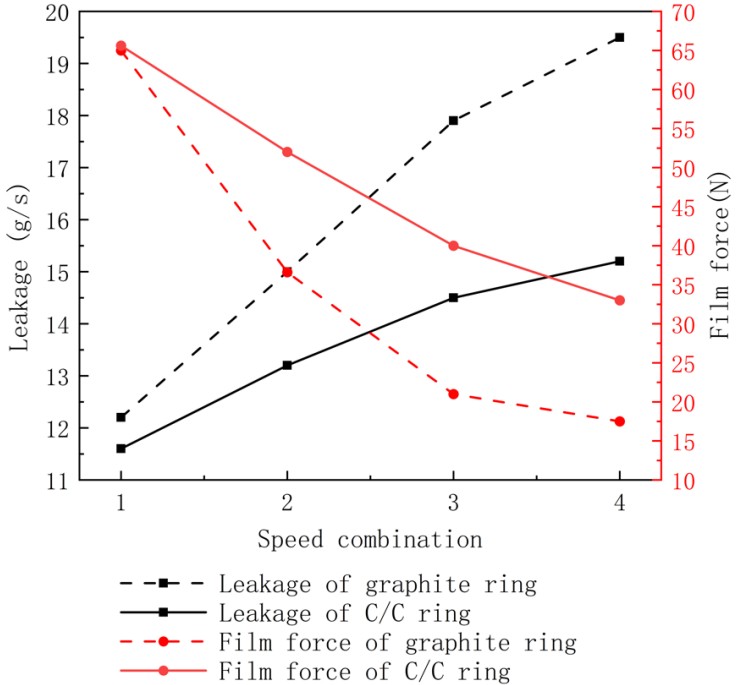

**Figure 17.** Comparison of the influence of rotational speed on the sealing performance of two materials.

**Table 5.** Speed combination of inner and outer rotors (reverse rotation of inner and outer rotors).

| Speed Combination | Inner Rotor Speed (rpm) | Outer Rotor Speed (rpm) | Relative Surface Speed (m/s) |
|:---:|:---:|:---:|:---:|
| 1 | 8000 | 5000 | 105.36 |
| 2 | 12,000 | 9000 | 170.24 |
| 3 | 15,000 | 11,000 | 210.77 |
| 4 | 16,000 | 12,000 | 226.99 |

As the rotational speed of the rotor increases, the gas film force of the cylindrical reverse inter-shaft gas film seal of the C/C composite seal ring and the graphite seal ring decreases because the increased rotational speed of the rotor increases the centrifugal expansion of the seal ring and the sealing gap, leading to a weakening of the gas dynamic pressure effect. The reverse rotation of the inner and outer rotor also has a certain weakening effect on the aerodynamic pressure effect of the gas film, thereby reducing the gas film force. However, the gas film force of the cylindrical reverse inter-shaft gas film seal of the C/C composite sealing ring is greater than that of the graphite sealing ring because of the large elastic modulus of the composite material used for the outer ring of the C/C composite sealing ring, which will hinder the centrifugal expansion of the inner ring when rotating at high speed, and because of the smaller sealing gap compared to the graphite sealing ring, resulting in a stronger aerodynamic pressure effect. Therefore, the gas film force of the cylindrical reverse axial gas film seal of the C/C composite seal ring is greater than that of the graphite seal ring.

## 6. Conclusions

Using a dual-layer C/C composite material with different elastic moduli (the outer layer stiffness is large, and the inner stiffness is small) to the reverse inter-shaft gas film seal structure can reduce centrifugal expansion and alleviate friction and wear issues, which is beneficial to achieve a balance between seal leakage and friction power consumption to some extent. Compared to a single material seal ring, a dual-layer sealing ring is conducive to improving the overall comprehensive performance of the seal by further structural optimization.

For a dual layer C/C composite sealing ring of a cylindrical reverse inter-shaft gas seal, by increasing the warp density of the inner sealing ring from 22 to 30, the leakage decreases by about 1.5%, and the friction power consumption increases by about 1.7%. By increasing the weft density of the inner sealing ring from 4 to 14, the leakage decreases by about 2.2%, and the friction power consumption increases by about 2.0%.

As the pressure difference increases, both the leakage and the gas film force increase. When the pressure difference increases from 20 kPa to 160 kPa, the leakage increases by 2.75 times, and the gas film force increases by 3.3 times. When the speed difference between the inner and outer rotors is maintained at 3000 rpm to 4000 rpm, and the rotational speed of the inner and outer rotors is increased by about one time simultaneously, the leakage increases by about 30%, and the gas film force decreases by about 49%.

Compared with a graphite sealing ring cylindrical reverse inter-shaft gas film seal, a C/C composite reverse inter-shaft gas film seal has less leakage and a larger gas film force under the same working conditions, especially at higher pressure differences or rotational speed. When the pressure difference is 160 kPa, the leakage of the C/C composite seal is 23% lower than that of the graphite ring seal, and the gas film force is 85% higher than that of the graphite ring. When the inner rotor rotational speed is 16,000 rpm, and the outer rotor rotational speed is 12,000 rpm, the leakage of C/C composite seal is 22% lower than that of graphite ring seal, and the gas film force is 94% higher than that of graphite ring seal.

**Author Contributions:** Conceptualization, H.S., S.Z. and X.Y.; methodology, H.S. and S.Z.; software, S.Z. and X.Y.; validation, H.S. and S.Z.; formal analysis, H.S. and S.Z.; resources, H.S.; writing—original draft preparation, H.S.; writing—review and editing, H.S. and S.Z.; visualization, S.Z.; project administration, H.S.; funding acquisition, H.S. All authors have read and agreed to the published version of the manuscript.

**Funding:** The authors would like to extend their appreciation to the National Natural Science Foundation of China for supporting this work under Grant 51575445.

**Data Availability Statement:** Not applicable.

**Conflicts of Interest:** The authors declare no conflict of interest.

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
