# Peer review of "Performance Analysis of the C/C Composite Cylindrical Reverse Inter-Shaft Gas Film Seal"

_lubricants, doi:10.3390/lubricants11050214_

Round 1

Reviewer 1 Report

The paper studies the performance of a C/C composite cylindrical reverse inter-shaft floating ring seal design. It is necessary to make corrections and answer the following questions:

1. In the Introduction Section, it is necessary to indicate the results of the study of floating ring seals, for example:

Arghir M. (2015). Experimental study of floating ring annular seals using high-speed optical techniques and mark tracking methods. CFM 2015 - 22ème Congrès Français de Mécanique. Lyon, France.

Zahorulko, A., Borsuk, S., Peczkis, G. (2022). Computational analysis of sealing and stability of a deformable floating and fixed rings of an annular seal. Journal of Engineering Sciences, Vol. 9(1), pp. D20-D29, doi: 10.21272/jes.2022.9(1).d4.

G. LI, Q. ZHANG, E. HUANG, Z. LEI, H. WU, G. XU (2019). Leakage performance of floating ring seal in cold/hot state for aero-engine, Chinese J. Aeronaut. 32 (2019) 2085–2094. doi:10.1016/j.cja.2019.03.004.

2. In the paragraph on page 2, lines 82-93, it is necessary to number the seal components shown in Figure 1.

3. Increase small symbols in Table 1 and text - lines 114, 115, 120, 144, 145, 171, 177, 181.

4. Why are the governing equations of laminar gas flow and sealing ring dynamics called analytical model? Moreover, these equations are further solved numerically. Why is the equation of the gas state not given?

5. Have the sealing ring dynamics (eq. 11, 12) and the sealing film thermal state (eq. 10) been analyzed? [M] is the sealing ring quality or mass matrix (line 162)?

6. Describe the symbol {sigma} (eq. 15).

7. Is the term opening force correct?

8. Why is the grid along the length of the gas seal film not shown (Figure 3)?

9. Show in the figures the results of the stress-strain state of the sealing ring and the gas film height (gap) and pressure distributions along the seal length L. Show the trajectories of gas flow due to the opposite rotation of the inner and outer rotors.

10. How was the balance between seal leakage and frictional power consumption determined?

11. Check the References. For example, Reference 13 incorrectly listed the journal quartile.

Reviewer 2 Report

See attached pdf with the reviewers' annotations. Paper relies too much on recent publications (Chinese origin). The topic was studied decades ago in the Western countries. Authors must demonstrate the need to use CFD/ANSYS (fluid solid interaction) for such a simple subject. 

Round 2

Reviewer 1 Report

After the revision, the paper may be published in the Lubricants journal. Please improve the quality of Figures 4 b, 5 b, and 8. If possible, represent the gas film pressure and clearance distribution along the axial width of the sealing ring on the graph.

Reviewer 2 Report

Revised version incorporates reviewer's recommendations. Thanks. However, the English language delivery remains rather poor. Extensive semantics revisions will be needed.
